The value of diffusion tensor tractography delineating corticospinal tract in glioma in rat: validation via correlation histology

http://orcid.org/0000-0002-4552-5682 Jia Xiaoxiong 1 2
Su Zhiyong 3
Hu Junlin 4
Xia Hechun 1 2 xhechun@nyfy.com.cn
Ma Hui 1
Wang Xiaodong 5
Yan Jiangshu 1
Ma Dede 6
1 Neurosurgery, General Hospital of NingXia Medical University , Yinchuan , China
2 Incubation Base of National Key Laboratory for Cerebrocranial Diseases, Ningxia Medical University , Yinchuan , China
3 Neurosurgery, Shouguang Traditional Chinese Medicine Hospital , Shouguang , China
4 Neurosurgery, Zigong Third People Hospital , Zigong , China
5 Radiology, General Hospital of NingXia Medical University , Yinchuan , China
6 Ningxia Medical University , Yinchuan , China
Abdullah Jafri
Electronic publication date: 2019 Feb 13
Publication date: 2019
Volume: 7
Electronic Location ID: e6453
Received 2018 Oct 2; Accepted 2019 Jan 15
Copyright: © 2019 Jia et al.
Copyright year: 2019
Copyright holder: Jia et al.
License: This is an open access article distributed under the terms of the Creative Commons Attribution License, which permits unrestricted use, distribution, reproduction and adaptation in any medium and for any purpose provided that it is properly attributed. For attribution, the original author(s), title, publication source (PeerJ) and either DOI or URL of the article must be cited.
License URL: https://creativecommons.org/licenses/by/4.0/

Keywords: Gliomas, Diffusion tensor imaging, Rats, Corticospinal tracts

Funding: National Natural Science Foundation of China 81260373 General Science Foundation of Ningxia Medical University 2015 XM2015082 This work was funded by the National Natural Science Foundation of China (Project No. 81260373) and the General Science Foundation of Ningxia Medical University 2015 (Project No. XM2015082). There was no additional external funding received for this study. The funders had no role in study design, data collection and analysis, decision to publish, or preparation of the manuscript.

==============================
Background

An assessment of the degree of white matter tract injury is important in neurosurgical planning for patients with gliomas. The main objective of this study was to assess the injury grade of the corticospinal tract (CST) in rats with glioma using diffusion tensor imaging (DTI).

Methods

A total 17 rats underwent 7.0T MRI on day 10 after tumor implantation. The apparent diffusion coefficient (ADC) and fractional anisotropy (FA) were acquired in the tumor, peritumoral and contralateral areas, and the ADC ratio (ipsilateral ADC/contralateral ADC) and rFA (relative FA = ipsilateral FA/contralateral FA) in the peritumoral areas were measured. The CST injury was divided into three grades and delineated by diffusion tensor tractography reconstruction imaging. The fiber density index (FDi) of the ipsilateral and contralateral CST and rFDi (relative FDi = ipsilateral FDi/contralateral FDi) in the peritumoral areas were measured. After the mice were sacrificed, the invasion of glioma cells and fraction of proliferating cells were observed by hematoxylin-eosin and Ki67 staining in the tumor and peritumoral areas. The correlations among the pathology results, CST injury grade and DTI parameter values were calculated using a Spearman correlation analysis. One-way analysis of variance was performed to compare the different CST injury grade by the rFA, rFDi and ADC ratio values.

Results

The tumor cells and proliferation index were positively correlated with the CST injury grade (r = 0.8857, 0.9233, P < 0.001). A negative correlation was demonstrated between the tumor cells and the rFA and rFDi values in the peritumoral areas (r = −0.8571, −0.5588), and the proliferation index was negatively correlated with the rFA and rFDi values (r = −0.8571, −0.5588), while the ADC ratio was not correlated with the tumor cells or proliferation index. The rFA values between the CST injury grades (1 and 3, 2 and 3) and the rFDi values in grades 1 and 3 significantly differed (P < 0.05).

Conclusions

Diffusion tensor imaging may be used to quantify the injury degrees of CST involving brain glioma in rats. Our data suggest that these quantitative parameters may be used to enhance the efficiency of delineating the relationship between fiber tracts and malignant tumor.

Introduction

Gliomas are characterized by invasive growth along fiber tracts in the white matter (Chen, Shi & Song, 2010). In particular, gliomas involving fiber tracts, such as corticospinal tract (CST), may result in neurological deficits due to the disruption, displacement or deformation of the fibers (Laundre et al., 2005). Therefore, delineating the different injury degrees in the CST should provide important additional information for better neurological treatment planning.

Diffusion tensor imaging (DTI) has been proven to classify damage to the fiber tracts and is used to quantitatively analyze the infiltration degree of the CST (Hervey-Jumper & Berger, 2014; Jeong et al., 2015). Diffusion tensor tractography (DTT) is a noninvasive tool that can be used to visualize major white matter tracts in three dimensions by setting the region of interest (ROI) as the fiber path (Conti et al., 2014; Forster et al., 2015; Jeong et al., 2014). Recently, fiber density index (FDi) values have been used to indicate the fiber density within the bundle passing through a unit volume, which can provide useful information for the assessment of the tumor border zone and peritumoral regions (Roberts et al., 2005; Stadlbauer et al., 2012). Some authors (Painter & Hillen, 2013; Witwer et al., 2002; Yu et al., 2005) have demonstrated that fractional anisotropy (FA) decreases in the white matter close to brain tumors, while apparent diffusion coefficient (ADC) increases in the tumor core and peritumoral areas. Rosenstock et al. (2017) showed that the FA value were lower and the ADC value were higher in peritumoral areas, which indicated that the FA and ADC value were affected at the peritumoral level. Additionally, tumor cell proliferation and invasion could result in a disruption of CST, and the degree of injury may reflect its biological behavior (Gao et al., 2017; Painter & Hillen, 2013).

Previous studies have shown that DTI can be used to assess the degree of damage in fiber tracts, and DTI has been used in neurological examinations of patients. However, to the best of our knowledge, whether the rFA, rFDi and ADC ratio, which are hypothesized as quantitative parameters in DTI, can be used to assess the injury grade in the CST or are correlated with pathology results in rats with glioma has not been established.

Therefore, in this study, the rFA, rFDi and ADC ratio were measured and evaluated in ROIs in the white matter in peritumoral areas. Furthermore, different degrees of CST injury were reconstructed and delineated by DTT. Importantly, in rats with glioma, the correlation between the gliomas and the damage to the CST was addressed by describing the DTI parameters in terms of the invasion and proliferation of the tumor cells in the peritumoral areas.

Materials and Methods

Rats and glioma cell injection

The animal studies were performed using protocols approved by the General Hospital of Ningxia Medical University Animal Care Committee (#2013-81260373). A total of 17 adult standard deviation (SD) rats (200–250 g in body weight) were used in this study. Rat glioma cells (C6) were cultured in Dulbecco’s modified Eagle’s medium supplemented with 10% fetal bovine serum. The procedures used to engraft the C6 tumors in the rat brains have been reported in detail in previous studies involving rat glioma models (Asanuma et al., 2008; Van Den Berge et al., 2015; Zhanfeng et al., 2015). In total, 17 glioma-bearing rats were used. For the implantations, each animal was anesthetized using 0.5% sodium pentobarbital (one mL/kg, intraperitoneal), and a suspension of C6 cells (one μL/min, 10 μL) was stereotactically injected into the cortex at a depth of five mm. The surgeries were performed under sterile conditions.

MRI protocol

All imaging was performed at 7.0T using a horizontal bore BRUKER Biospec 70/20 Advance MRI instrument (Bruker, Karlsruhe, Germany) at the National Center for Nanoscience and Technology in Beijing, China. The radiofrequency coil used was a surface coil. The rats were anesthetized using 0.5% sodium pentobarbital (one mL/kg, intraperitoneal) and then placed into a head holder with a bite-bar and ear bars to immobilize the head. The body temperature was maintained at 37.5 °C using a hot water-circulating bath. The MRI scans were performed 10 days after the cell implantation.

The following T2-weighted imaging parameters were used to obtain the images: repetition time (TR) = 2,000 ms, echo time (TE) = 20 ms, field of vision (FOV) = 40 × 40 mm2, slice thickness = 1.0 mm, acquisition matrix = 256 × 256, NEX = 2, flip angle = 90°, and number of slices = 20. The following DTI parameters were used to obtain the images: TR = 5,000 ms, TE = 27 ms, FOV = 40 mm × 40 mm2, slice thickness = 1.0 mm, acquisition matrix = 128 × 128, flip angle = 90°, number of slices = 20, and number of gradients = 25. After the image acquisition, the data were transferred to a MATLAB workstation for analysis.

ROI measurements

The FA, ADC and FDi values were measured from ROIs chosen by two experienced neuroradiologists (Dr. Wang and Dr. Huang) with at least 7 years of work experience (Price et al., 2003). The FA, ADC and FDi values were calculated in the following ROIs, which were both contralateral and ipsilateral to the tumor: in the tumor and peritumoral areas. To enhance the accuracy of the measurement, the ROIs were selected based on five different brain spots in each area (Blasiak et al., 2010; Lu et al., 2003). The FA, ADC and FDi value measurements in the contralateral and ipsilateral peritumoral areas were used to calculate the ADC ratio (ipsilateral ADC/contralateral ADC), rFA (relative FA = ipsilateral FA/contralateral FA) and rFDi (relative FDi = ipsilateral FDi/contralateral FDi).

CST injury grade

The CST originates in the motor cortex area and then travels through the posterior limb of the internal capsule to enter the cerebral peduncle. Therefore, the ROIs of the CST were usually placed in the subcortical white matter of the precentral gyrus (Brodmann 4) and premotor cortex (Brodmann 6) area (ROI), the posterior limb of the internal capsule (ROI), or the entire cerebral peduncle (ROI). To isolate fibers of the CST with the “AND” operation of DTT software, the traces that passed through these ROIs (two or three ROIs) were considered to be components of the CST (Holodny et al., 2005; Suzuki et al., 2009). During fiber tracking by DTT, the DTI data were processed and analyzed off-line using the TrackVis software package and a FA value of 0.20. In our study, the motor cortex area (initial ROI) and cerebral peduncle (target ROI) were chosen by two experienced neuroradiologists (XDW, XYH).

In this study, the white matter tracts in 17 glioma-bearing rats were classified according to the criteria of displacement, infiltration and disruption (Gao et al., 2017; Witwer et al., 2002) as follows: (1) grade 1: displaced and edematous if the tracts displayed a normal anisotropy signal relative to the corresponding tracts in the contralateral hemisphere, displacement of the fibers by the tumor and some peritumoral edema delineated by the DTT imaging (Fig. S1). (2) grade 2: infiltrated if the tracts showed decreased anisotropy but remained intact without significant disruption; (3) grade 3: disrupted if anisotropy was markedly reduced such that the tract could not be identified on the FA map, and interruption of the white matter tracts can be observed on DTT imaging.

Animal test

The animal test was performed one day before the MRI scan by two observers (Dr. H and Dr. S) in a blinded fashion. Two tests (modified from Garcia) were used to evaluate the motor function of the rats (Garcia et al., 1995), and the scores ranged from one (minimum) to six (maximum). The grading scale for motor deficit (modified from Bederson) was divided into two grades (moderate: scores 3–6, and severe: scores 1–2) as described below (Bederson et al., 1986).

Spontaneous activity: The rat was observed for 2 min in a transparent plastic cube 30 cm in length. Climbing test: The animal was placed and climbed the wire mesh wall. The rat’s tail was held, and the rat was pulled off, during which the strength of the rat’s four limbs was noted.

Histology and immunohistochemistry

After completing the MRI experiments, the rats were sacrificed 10 days after implantation, and the whole brains of the rats were removed from the skull and stained using hematoxylin-eosin (H&E) and Ki67 (Del Duca, Werbowetski & Del Maestro, 2004). The fraction of glioma invasion cells among the total number of cells in the peritumoral areas was assessed in H&E stained sections in a high-powered field (HPF), and the regions of measurement were selected based on five different brain spots in each area. Using Ki67 staining, we evaluated the fraction of proliferating cells in each peritumoral regions and selected five different brain spots (Smith et al., 2017). The histological specimens were analyzed to determine the regions that corresponded to the quantitative MRI measurements (Wang & Zhou, 2012). These tumor cells were compared with the data obtained using the DTI. The maps used to obtain the microscopic cell counts were derived using the Image-Pro Plus software package.

Statistical analysis

The statistical calculations were performed using the GraphPad Prism 5 software package and the statistical package SPSS (Version 21, SPSS). The FA, ADC and FDi data are presented as the mean ± SD as indicated. A paired-sample t-test was applied to analyze the significance of the differences observed in the parameters between the tumoral and peritumoral areas, between the contralateral normal area and tumoral areas, and between the contralateral normal area and peritumoral areas. A P-value less than 0.05 was considered to indicate a significant difference. The correlations between the CST injury grades and the rFA and rFDi in the tumor cells were calculated using a Spearman correlation analysis. The rFA and rFDi values were compared among groups with different CST injury degrees with a one-way analysis of variance.

Results

Summary of quantitative parameters and CST injury grades in 17 rats

The mean ± SD of the FA, ADC and FDi were measured in the tumoral and peritumoral areas and contralateral hemisphere tissue (Table S1 and S3; Figs. 1A–1C). The DTI analysis and histology results of the peritumoral areas are shown in Table S2.

Figure 1 Comparisons of the FA, ADC and FDi values.

Comparisons of the FA (A), ADC (B) and FDi (C) values in different brain regions and correlation between the DTI quantitative parameters (rFA, rFDi and ADC ratio) and histological results of peritumoral areas. (D, E), A negative correlation is observed among the tumor cells, proliferation index and rFA and rFDi values in the peritumoral areas, while the ADC ratio was not correlated with the tumor cells and proliferation index. Tumor cells (%) = density of glioma invasion cells present in the peritumoral areas per high powered field; Proliferation index (%) = the fraction of proliferating cells in each peritumoral region; TA, tumoral areas; PA, peritumoral areas; Con, contralateral. Values are mean ± SD. n = 17. *P < 0.05, **P < 0.01 and ***P < 0.001.

The tumor cells and proliferation index were positively correlated with the CST injury grade (r = 0.8857, 0.9233, P < 0.001). A negative correlation was observed between the tumor cells and the rFA and rFDi values in the peritumoral areas (r = −0.8571, −0.5588, respectively), and the proliferation index was negatively correlated with the rFA and rFDi values (r = −0.8698, −0.5856, respectively), while the ADC ratio was not correlated with the tumor cells or proliferation index (Figs. 1D and 1E).

Diffusion tensor imaging analysis and CST injury grades

The rFA values significantly differed between the CST injury grades (1 and 3, 2 and 3) (P < 0.05) (Fig. 2A); the rFDi values significantly differed between the grades (1 and 3) (P < 0.05) (Fig. 2B), while the ADC ratio (Fig. 2C) and rFA and rFDi values in the remaining CST injury grade groups did not significantly differ.

Figure 2 Comparison of the rFA, rFDi and ADC ratio.

Comparison of the rFA (A), rFDi (B) and ADC ratio (C) among different CST injury grades in peritumoral areas. Grade 1 (black), Grade 2 (red), Grade 3 (blue); rFA = ipsilateral FA/contralateral FA; rFDi = ipsilateral FDi/contralateral FDi; ADC ratio = ipsilateral ADC/contralateral ADC; n = 17. *P < 0.05, and ***P < 0.001.

Histology and immunohistochemistry of peritumoral areas

The tumor cells in the peritumoral areas were assessed in H&E stained sections and compared among the different CST injury grade groups of 17 rats. Grade 1 (Fig. 3A) had a median value of 39% per HPF under 40 × magnification; grade 2 (Fig. 3B) had a median of 48% per HPF, and grade 3 (Fig. 3C) had a median of 57% per HPF. The tumor cells significantly differed between the CST injury grade groups (1 and 2, 1 and 3, 2 and 3) (Fig. 3D).

Figure 3 Three CST injury grades show different infiltration degrees and growth rates in peritumoral areas.

(A–C) H&E sections were used to assess the percentage of tumor cells among the total number of cells per high powered field under 40 × magnification. (A) Grade 1 samples had a median tumor cell fraction of 39%. (B) Grade 2 samples had a median tumor cell fraction of 48%. (C) Grade 3 samples had a median tumor cell fraction of 57%. (D) Comparison of tumor cells among different CST injury grades in peritumoral areas. (E–G) Ki67 immunohistochemistry staining under 40 × magnification to record the positive fraction in peritumoral areas. (E) Grade 1 samples had a median Ki67 positive cell fraction of 16%. (F) Grade 2 samples had a median Ki67 positive cell fraction of 24%. (G) Grade 3 samples had a median Ki67 positive cell fraction of 44%. (H) Comparison of the proliferation index among different CST injury grades in peritumoral areas. (D, H) Grade 1 (black), Grade 2 (red), Grade 3 (blue); n = 17. *P < 0.05, **P < 0.01 and ***P < 0.001.

The proliferation index in the peritumoral areas was assessed and compared among the different CST injury grade groups by using Ki67 immunohistochemistry staining. In grade 1 (Fig. 3E), the fraction of Ki67 positive cells was 16% under 40 × magnification; in grade 2 (Fig. 3F), the median was 24%, and in grade 3 (Fig. 3G), the median was 44%. The proliferation index significantly differed among the grades (1 and 3 and 2 and 3) (P < 0.05), while in the remaining groups (1 and 2), no significant differences were observed (Fig. 3H).

Displacement of corticospinal fibers

The T2WI and DTI scans were performed in 17 glioma-bearing rats. On the conventional T2WI, the tumors showed irregular mixed signals in regions involved with the right motor cortex (Fig. 4A). On the FA maps, normal diffusion anisotropy signals were observed in the right hemisphere relative to those in the contralateral areas (Figs. 4B and 4C). In the DTT, the initial point was the motor cortex area, and the cerebral peduncle was the seed point. Furthermore, the 3D reconstruction map showed that the right CST was displaced anterolaterally by the mass lesion compared with the left fibers (Fig. 4D).

Figure 4 Displacement of corticospinal tract (CST) with CST injury grade 2.

(A). T2W image showing a tumor with a mixed signal in the vicinity of the right motor cortex. (B and C) Representative FA color imaging and FA imaging. The anisotropy signal decreased compared with that in the contralateral hemisphere. Green, anterior/posterior; blue, inferior/superior; and red, right/left. (D) Representative 3D reconstruction DTT imaging; the right CST is displaced anterolaterally by the mass lesion (light blue irregular solid shape) compared with the left fibers; right motor cortex (red solid sphere), left normal anatomical motor cortex (dark blue sphere), right cerebral peduncle (yellow sphere), left cerebral peduncle (green sphere); R = right side.

Disruption of corticospinal fibers

As shown in Fig. 5, the corresponding fiber tracts were disrupted by the tumors using 3D reconstruction DTT imaging. The conventional MRI scans revealed a large hypo-intense lesion on T2W imaging involving the right motor area and subcortical regions (Fig. 5A). On the FA color maps, the anisotropy in the ipsilateral hemisphere was markedly reduced compared with that in the left hemisphere such that the fibers could not be tracked (Fig. 5B). Figure 5C shows the DTT reconstructed complex fibers without the target and initial ROIs in the superior view. Compared with the contralateral hemisphere, the fibers on the right side were disrupted and affected by the tumor (light yellow areas). Green, anterior/posterior; blue, inferior/superior; and red, right/left. The reconstruction of the DTT images showed that the right CST was no longer present compared with the contralateral fibers in the left hemisphere (Fig. 5D).

Figure 5 Disruption of corticospinal tract (CST) with CST injury grade 3.

(A) T2W image showing a large tumor with necrosis in the right hemisphere. (B) Representative FA color imaging. Anisotropy was markedly reduced. (C) Representative DTT reconstruction complex fibers without the target and initial regions of interest (ROIs) in the superior view. Green, anterior/posterior; blue, inferior/superior; and red, right/left. (D) Representative 3D DTT imaging; the corresponding CST fibers are no longer present in their normal anatomical location within the right motor area involving the tumor (light yellow areas); right motor cortex (red irregular area), left motor cortex (light blue area), right cerebral peduncle (yellow sphere), left cerebral peduncle (orange sphere). R = right side.

Discussion

Resection of gliomas involving motor functional areas requires a detailed understanding of the relationships between the motor cortex and surrounding brain tissue and subcortical white matter fibers (Gao et al., 2017; Hervey-Jumper & Berger, 2014). The aim of this study was to quantitatively analyze the degree of damage to the fibers affected by gliomas and evaluate the role of DTI in visualizing the CST in the vicinity of the motor cortex.

Diffusion tensor imaging parameters are used to evaluate different grades of gliomas, assess the extent of peritumoral edema and reflect the degree of malignant tumor infiltration (Budde et al., 2011; Harsan et al., 2010; Hou et al., 2018). Our statistical results showed that the differences in the FA, ADC and FDi between the tumor areas and peritumoral areas are significant and that the rFA and rFDi values were negatively correlated with the tumor cells and proliferation index in the peritumoral areas, while the ADC ratio values did not significant differ. Using H&E and Ki67 staining, in this study, the higher number of tumor cells and proliferative index in the peritumoral areas indicate high invasion into the white matter. In our experience, the FA and FDi values decrease in peritumoral areas likely due to tumor infiltration and proliferation. Consequently, invasion of tumor cells could lead to damage to fibers in gliomas.

Diffusion tensor imaging can indicate whether anatomically intact fibers are present in abnormally appearing areas of the brain and characterize the integrity of white matter tracts in patients with brain tumors (Stadlbauer et al., 2012; Witwer et al., 2002). In this study, we demonstrated that the rFA and rFDi values significantly differed based on the grade of CST damage. Some studies have found that the FA value is related to the integrity of subcortical fiber tracts (Harsan et al., 2010). (Chen, Shi & Song, 2010) found that the FDi could be used to evaluate fiber disruption and tumor infiltration. White matter tracts were classified and predictive of total resection or partial resection (Jia et al., 2013). The degree of CST injury from glioma was correlated with the rFA and rFDi values, suggesting that DTI can reflect different degrees of injury from glioma in CST area and predict more useful information regarding the relationship between the nerve fibers and the lesions. Therefore, DTI parameters could be used to evaluate peritumoral areas along the path of CST fibers that have been affected by tumors.

The goal of charactering the CST injury grades and delineating the different components of the fiber tracts is to provide more accurate information to describe the spatial relationship between a malignant tumor and normal tissues. In our study, we showed that the tracking of the CST could be performed, visualized and distinguished by DTT reconstruction imaging. DTT can be used to visualize major white matter tracts by setting the ROI on the path of the fibers using anatomic structures (Baldoli et al., 2015; Kleiser et al., 2010). In our study, we verified the efficacy of a protocol that uses initial and target ROIs to delineate CST involved in diffusely infiltrating gliomas in rats. Better delineation of the CST injury grade could provide important additional information for tumor removal assessment.

Although we evaluated the fiber tracts by DTI parameters compared with histology, we suggest performing experiments to delineate white matter using additional staining. The further development of our findings could expand the range of these applications beyond the limit of the motor areas and increase the number of animal models.

Conclusions

In conclusion, our results demonstrate that DTI may quantitatively evaluate the injury degrees in fiber tracts involving brain gliomas in rats. Our data suggest that these quantitative parameters may be used to enhance the efficiency of delineating the relationship between fiber tracts and tumors.

Supplemental Information

Supplemental Information 1 MRI raw data.

Click here for additional data file.

Supplemental Information 2 MRI raw data: FA, ADC, FDi.

Click here for additional data file.

Supplemental Information 3 FA, ADC and FDi values for tumoral and peritumoral areas and contralateral areas.

Note: FA = fractional anisotropy; ADC = apparent diffusion coefficient; FDi = fiber density index; TA = tumoral areas; PA = peritumoral areas; Con = contralateral.

Click here for additional data file.

Supplemental Information 4 Histological measures, CST injury grades and animal test scores in 17 rats and a summary of the MRI parameters in the peritumoral areas.

Note: rFA = ipsilateral FA/contralateral FA; rFDi = ipsilateral FDi/contralateral FDi; ADC ratio = ipsilateral ADC/contralateral ADC; Tumor cells(%) = percentage of glioma cells among the total number of cells in the peritumoral areas per high-powered field; Proliferation index(%) = fraction of proliferating cells in each peritumoral region; CST = corticospinal tract; Moderate (M), Severe (S). Three rats exhibited grade 1 injury, six rats demonstrated grade 2 injury, and eight rats displayed grade 3 injury.

Click here for additional data file.

Supplemental Information 5 Comparison of the FA, ADC and FDi values among tumoral and peritumoral areas and contralateral areas.

Note: FA = fractional anisotropy; ADC = apparent diffusion coefficient; FDi = fiber density index; TA = tumoral areas; PA = peritumoral areas; Con = contralateral;.

Click here for additional data file.

Supplemental Information 6 Corticospinal tract (CST) with CST injury grade 1.

A, T2W image showing a tumor (T indicates the tumor). B, Representative FA color imaging. The anisotropy signal displayed a slight decrease relative to the corresponding tracts in the left hemisphere. C, Representative DTT imaging, top view. The right CST is slightly displaced by the lesion; R = right side.

Click here for additional data file.

The authors are grateful to Enshan Han and Xiaoli Liu (Neuropathologist Department, General Hospital of NingXia Medical University, Yinchuan, Ningxia, China) for performing the histological examination.

Additional Information and Declarations

Competing Interests

Author Contributions

Animal Ethics

Data Availability

The authors declare that they have no competing interests.

Xiaoxiong Jia conceived and designed the experiments, performed the experiments, analyzed the data, contributed reagents/materials/analysis tools, prepared figures and/or tables, authored or reviewed drafts of the paper, approved the final draft.

Zhiyong Su performed the experiments, prepared figures and/or tables.

Junlin Hu performed the experiments, prepared figures and/or tables.

Hechun Xia conceived and designed the experiments, contributed reagents/materials/analysis tools, authored or reviewed drafts of the paper, approved the final draft.

Hui Ma contributed reagents/materials/analysis tools.

Xiaodong Wang analyzed the data.

Jiangshu Yan contributed reagents/materials/analysis tools.

Dede Ma analyzed the data, prepared figures and/or tables.

The following information was supplied relating to ethical approvals (i.e., approving body and any reference numbers):

Animal Ethics Committee of the General Hospital of Ningxia Medical provided full approval for this animal research (#2013-81260373).

The following information was supplied regarding data availability:

The raw data are available in the Supplementary Materials.

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
