# Peer review of "The value of diffusion tensor tractography delineating corticospinal tract in glioma in rat: validation via correlation histology"

_PeerJ, doi:10.7717/peerj.6453_

## Round 0.1 · original submission · Major Revisions

Dear Authors,

There are many technical and methodology flaws that need to be looked into for the sake of good science highlighted by the two peer reviewers,i do think if you are unable to do the major revision the manuscript may not be able to be accepted by PeerJ.

Reviewer 1 ·

Basic reporting

Authors do not discuss other difusion MRI methods besides DTI.

No discussion of the effects of edema. Was edema evaluated either by imaging or by histologic examination?

There are no behavioral tests to establish that the “disruption of tracts” seen by imaging and histology had expected effect on the motor function of the rats.

Methods do not clearly describe how CST was selsected. (line 220-221) In the DTT, the initial point was the motor cortex area, and the cerebral peduncle 221 was the seed point. How was motor cortex defined? Where in the cerebral peduncle

How were the sites of histologic evaluation selected? How were the registered to the sites of DTI measurements? How can the authors know that these measurements are representative of the same region as the DTI measurements? It seems as if only 5 HPF were examined per rat. How were these selected?

The histologic examination does not investigate tract integrity per se. There are no myelin stains or other efforts to examine whether the tracts are in continuity or not.



Given that the motor function of the animals was not assessed, the histologic integrity of the tracts themselves were not addressed, and the colocalization o the imaging and histologic findings were not described, the conclusion is not fully supported by the findings.
In conclusion, our results demonstrate that DTI imaging may quantitatively evaluate the injury degrees in fiber tracts involving brain gliomas in rats. Our data suggest that these quantitative parameters may be used to enhance the efficiency of delineating the 284 relationship between fiber tracts and tumors.


Minor points:

Line 85: degree of damage rather than damage degree

Line 219 “right” hemisphere is not clear from methods whether this is the injected or non-injhected hemisphere.

Line 239 should be affected not effected

Line 260 unclear awkward phrasing “were classified and predictive of total resection or partial resection”

Error in line 266: charactering


Unclear line 224
“the corresponding fiber tracts were disrupted by the tumors using 3D reconstruction DTT imaging”


Results section is very dense and hence difficult to read and follow. This section could benefit from introductory and concluding sentences .

Experimental design

Authors do not discuss other difusion MRI methods besides DTI.

No discussion of the effects of edema. Was edema evaluated either by imaging or by histologic examination?

There are no behavioral tests to establish that the “disruption of tracts” seen by imaging and histology had expected effect on the motor function of the rats.

Validity of the findings

Given that the motor function of the animals was not assessed, the histologic integrity of the tracts themselves were not addressed, and the colocalization o the imaging and histologic findings were not described, the conclusion is not fully supported by the findings.

·

Basic reporting

Language: I am not a native Speaker. For that reason, I am not able to proof the used language. I did not note any serious error.

Background / context: The used literature helped to follow the results and supported sufficiently the discussion. The publication from Berlin should be discussed because there were found similar finding like in this study in brain Tumor patients (Specific DTI seeding and diffusivity-analysis improve the quality and prognostic value of TMS-based deterministic DTI of the pyramidal tract).

Structure: article is well structured. Some Images of the figures seem to be very small and might be enlarged.

Experimental design

Visualization of all 3 grades of CST injury should be visualized (not only 2 grades). Additionally, there is a bias to compare the CST injury grade (which bases on the relative FA-value) with the rFA-value (which is calculated by the ipsilesional FA-value and the contralateral FA-value) - Fig 2.

Some data are repeated in the text, the tables and the figures. I also recommend to add p-values in the tables.

I am not able to proof the histological analysis.

Validity of the findings

To my Knowledge, this is the first study analyzing the diffusivity parameters in rats where Tumor cells were implemented. The shown results go in line with data published in patients.

---

## Round 0.2 · accepted · Accept

Dear Authors,

Your revised manuscript is accepted for publication.

Congratulations!

Reviewer 1 ·

Basic reporting

The report is acceptable for publication.

Experimental design

acceptable

Validity of the findings

acceptable